

# TICI: a taxon-independent community index for eDNA-based ecological health assessment

Shaun P. Wilkinson[1,2], Amy A. Gault[1], Susan A. Welsh[1], Joshua P. Smith[3,4], Bruno O. David[4], Andy S. Hicks[5,6], Daniel R. Fake[6], Alastair M. Suren[7], Megan R. Shaffer[8], Simon N. Jarman[2] and Michael Bunce[2,9,10]

[1] Wilderlab NZ Ltd., Wellington, New Zealand
[2] School of Molecular and Life Sciences, Curtin University, Bentley, Western Australia, Australia
[3] School of Science, The University of Waikato, Hamilton, Waikato, New Zealand
[4] Waikato Regional Council, Hamilton, Waikato, New Zealand
[5] Ministry for the Environment, Wellington, New Zealand
[6] Hawke's Bay Regional Council, Napier, Hawke's Bay, New Zealand
[7] Bay of Plenty Regional Council, Tauranga, Bay of Plenty, New Zealand
[8] School of Marine and Environmental Affairs, University of Washington, Seattle, WA, United States of America
[9] Department of Conservation, Wellington, New Zealand
[10] School of Biomedical Sciences, University of Otago, Dunedin, Otago, New Zealand

Corresponding author
Shaun P. Wilkinson,
shaun@wilderlab.co.nz

## ABSTRACT

Global biodiversity is declining at an ever-increasing rate. Yet effective policies to mitigate or reverse these declines require ecosystem condition data that are rarely available. Morphology-based bioassessment methods are difficult to scale, limited in scope, suffer prohibitive costs, require skilled taxonomists, and can be applied inconsistently between practitioners. Environmental DNA (eDNA) metabarcoding offers a powerful, reproducible and scalable solution that can survey across the tree-of-life with relatively low cost and minimal expertise for sample collection. However, there remains a need to condense the complex, multidimensional community information into simple, interpretable metrics of ecological health for environmental management purposes. We developed a riverine taxon-independent community index (TICI) that objectively assigns indicator values to amplicon sequence variants (ASVs), and significantly improves the statistical power and utility of eDNA-based bioassessments. The TICI model training step uses the Chessman iterative learning algorithm to assign health indicator scores to a large number of ASVs that are commonly encountered across a wide geographic range. New sites can then be evaluated for ecological health by averaging the indicator value of the ASVs present at the site. We trained a TICI model on an eDNA dataset from 53 well-studied riverine monitoring sites across New Zealand, each sampled with a high level of biological replication ($n = 16$). Eight short-amplicon metabarcoding assays were used to generate data from a broad taxonomic range, including bacteria, microeukaryotes, fungi, plants, and animals. Site-specific TICI scores were strongly correlated with historical stream condition scores from macroinvertebrate assessments (macroinvertebrate community index or MCI; $R^2 = 0.82$), and TICI variation between sample replicates was minimal (CV = 0.013). Taken together, this demonstrates the potential for taxon-independent eDNA analysis to

provide a reliable, robust and low-cost assessment of ecological health that is accessible to environmental managers, decision makers, and the wider community.

## INTRODUCTION

The overexploitation of natural resources and capital is driving severe global declines in habitat quality and biodiversity (*Turvey & Crees, 2019*; *Dasgupta, 2021*). High-profile extinctions and incursions receive widespread attention, but the gradual and less perceptible shift in overall ecological health poses a far greater threat (*Pauly, 1995*; *Soga & Gaston, 2018*). The normalizing of degraded habitats and diminished biodiversity can lead to a lack of public awareness and political will to take conservation action (*Worm et al., 2006*). Worse still, continued inaction on gradual ecological degradation can lead to irreversible tipping points (*Scheffer, 2020*; *Carrier-Belleau et al., 2022*). Effective, whole-ecosystem monitoring is important for recognising ecological declines and influencing mitigating actions. Baseline monitoring provides an objective reference point against which changes can be measured, enabling managers to understand how ecosystems have been altered and the extent of the damage. Monitoring and reporting on biodiversity can also deliver greater investment, more effective management practices, and ultimately, better conservation outcomes (*Rodrigues et al., 2004*). Ecological health metrics provide crucial indicators against which restoration progress can be measured over time. Improvements can be achieved through financial instruments that incentivise responsible and sustainable resource use such as biodiversity offset and credit schemes, and/or through higher levels of performance and motivation for achieving specific goals or targets (*Locke & Latham, 2002*). Well-formulated goals designed to achieve explicit, measurable biodiversity indicators will be critical for governments and society to halt and reverse these worsening trends (*Díaz et al., 2020*).

While there is a clear and urgent need to upscale ecological monitoring, a lack of objective, consistent and cost-effective tools hinder our ability to accurately track ecological trends (*Urban et al., 2021*). Most existing biological survey methods require extensive taxonomic expertise, expensive equipment, and are specific to one, or a few easily-identified taxon groups such as fish or insects, which may not always be the most responsive indicators of environmental stress (*Leese et al., 2018*). Environmental DNA (eDNA) metabarcoding offers a powerful biomonitoring solution that enables field practitioners to survey taxa across the tree-of-life with little training and expertise (*Stat et al., 2017*). This method involves a sample collection step (often filtration of source water to capture biological matter) followed by DNA extraction, PCR amplification of short but taxonomically-informative genetic markers, high-throughput sequencing (HTS), and taxon-assignment by mapping eDNA amplicon sequence variants (ASVs) against a reference database. eDNA metabarcoding offers many advantages over traditional morphology-based survey methods. These include: (1) increased sensitivity, with lower non-detection rates and

increased spatial coverage (*David et al., 2021*); (2) improved specificity, with the ability to distinguish cryptic or morphologically similar species (*Leese et al., 2018*); (3) expanded scope, with the ability to identify a much higher species richness per observation than can be surveyed visually, including species from taxa that are responsive indicators of ecosystem function such as bacteria, ciliates, diatoms and foraminifera (*Glasl, Webster & Bourne, 2017*; *Sagova-Mareckova et al., 2021*); (4) improved consistency, with the ability to standardize methods, and thus bridge the data-quality gap between professional and amateur or community-led monitoring; (5) major savings in cost, time and effort; (6) the opportunity to monitor wildlife in a non-invasive manner (*Hering et al., 2018*); and (7) sample and data reusability. In particular, archived samples can be re-analysed using new methods/assays, unidentified ASVs can be re-analysed in light of new reference sequence data, and accessible eDNA data can be used by different agencies for different purposes (*Berry et al., 2021*).

The many advantages of eDNA metabarcoding have prompted a rapid global uptake in recent years (*Takahashi et al., 2023*). Yet there remain several limitations that must be clearly communicated to avoid misinterpreting and misusing eDNA data (*Hering et al., 2018*). These include (1) false positives resulting from DNA contamination, either through sample mishandling or the introduction of DNA from non-living sources such as fecal matter from predators (*Jerde et al., 2011*); (2) false negatives resulting from biases in primer affinity for target ASVs that can lead to 'dropouts' of certain species or groups (*Clarke et al., 2014*); (3) decoupling between ASV counts and biomass or abundance of target organisms; and (4) incompleteness and inaccuracies in reference sequence databases, that can lead to misclassification and overclassification errors when assigning taxonomy (*Edgar, 2018*). Of the large numbers of ASVs generated by metabarcoding analysis, only a fraction can typically be confidently identified to species or genus level, and these tend to be biased toward charismatic fauna that may be over-represented in reference databases (*Kermarrec et al., 2013*).

The widespread uptake of eDNA metabarcoding for environmental monitoring is generating a vast amount of data (*Berry et al., 2021*). Interpreting this data for resource management or policy formulation is a significant challenge, since it requires expertise in the genomic methods that generate the data combined with ecological knowledge of the system under study. Decision makers can struggle to make sense of large lists of unfamiliar taxa, and must find ways to distil these complex datasets to meaningful metrics upon which good environmental decisions can be based. Biotic indices offer a solution to reduce complex multivariate community data to a single value. While such an approach may not tell the whole story, it helps to communicate environmental information to non-specialists and convey important patterns and trends for 'state and trend' reporting (*Pawlowski et al., 2018*). Key questions can then be addressed such as how a specific study site compares to other surveyed sites, and how the biological community has changed over time in response to external inputs, or a changing climate. The underlying assumption of biotic indices is that the biological community is a product of both environmental parameters and a unique set of stressors at a site. The biotic index thus provides a time-integrated picture of environmental factors.

Many biotic indices are derived by assigning indicator values to taxa across a continuum of tolerance or susceptibility to environmental stress. The selection of indicator taxa has historically been based on ease of detection and identification (*Rosenberg & Resh, 1993*; *Stark & Maxted, 2007*), but eDNA now offers access to a much wider suite of indicators, including bacteria and microeukaryotes that have traditionally been difficult to identify morphologically. eDNA-based biotic indices are increasingly being developed and applied for measuring ecological change, and many previously overlooked indicators have been shown to be sensitive barometers for ecological health across different habitat types (*Chariton et al., 2015*; *Laroche et al., 2016*; *Stoeck et al., 2018*; *DiBattista et al., 2020*). Yet most eDNA-based indices still rely on a taxon-assignment step, after which a large portion of the data is usually discarded if taxon identification is not possible with current reference data. More recently, eDNA-based indices have been developed without incurring data-loss through taxon-assignment, by assigning indicator values to amplicon sequence variants (ASVs) or operational taxonomic units (OTUs) rather than assigned taxa (*Cordier et al., 2021*; *Frühe et al., 2021*; *Lanzén et al., 2021*). These taxon-independent approaches can out-perform taxonomic indices, particularly for groups where reference data are limited, such as bacteria. For example, *Aylagas et al. (2021)* used a regression approach to assign indicator values to microbial ASVs along a sediment organic enrichment gradient near aquaculture farms, and the taxon-independent index outperformed the taxon-dependent microgAMBI index by improving the discrimination of changes in organic enrichment. To date, taxon-independent indices have been largely developed within individual taxon groups such as diatoms (*Apothéloz-Perret-Gentil et al., 2017*; *Feio et al., 2020*) and bacteria (*Aylagas et al., 2021*). With a renewed focus on integrated, whole-ecosystem monitoring, there remains a need to apply these taxon-independent approaches to a wide taxonomic range of indicator species to gain a more comprehensive picture of ecological change.

Biodiversity loss in New Zealand has been driven by decades of land conversion and intensive use for agriculture and urban development, coupled with the spread of exotic organisms. This has led to demonstrable declines in the ecological health of rivers, streams, and lakes (*Julian et al., 2017*). New legislation has been introduced to ensure that natural and physical resources are managed in a way that prioritizes the health and well-being of water bodies and freshwater ecosystems (National Policy Statement for Freshwater Management; *New Zealand Ministry for the Environment, 2020*). However, New Zealand's environmental reporting system remains badly fragmented (*New Zealand Parliamentary Commissioner for the Environment, 2019*) and there remains ongoing debate on the revised NPS-FM 2020 thresholds for describing many of the potentially anthropogenically-derived impacts of excessive nutrients and sediment on ecosystem health (*Canning, Joy & Death, 2021*). Methods for ecological health assessment in lotic systems have traditionally centred around two approaches: the use of electrofishing machines (EFM) or netting/trapping to survey fish communities (*Joy et al., 2013*); and kick-net sampling of macroinvertebrates in wadeable streams (*Stark et al., 2001*). EFM and netting data is used to compile an Index of Biotic Integrity (*Joy & Death, 2004*) which gives an indication of observed *versus* expected diversity in a given habitat, and can reveal issues arising from pollution and fish passage barriers. Macroinvertebrate samples are collected with a kick-net, preserved, and sent

to a number of different laboratories, where invertebrates identified to family or genus level where possible using common diagnostic keys (*Winterbourn, 1973*; *Winterbourn, Gregson & Dolphin, 2006*; *Chapman, Lewis & Winterbourn, 2011*). A number of indicator taxa were previously assigned indicator values based on their susceptibility or tolerance to organic enrichment using an iterative weighting procedure, and indicator values are averaged over the entire community to produce the MCI index (*Stark & Maxted, 2007*). A high MCI score (>120) indicates a healthy stream, while low scores (<90) generally indicate impacts associated with modified land-use, low vegetation cover, homogenized instream habitat, and terrestrial runoff (*Stark, 1985*; *Stark & Maxted, 2007*). Both the EFM/netting and macroinvertebrate sampling are restricted to wadeable streams and rely on morphological identification of taxa by skilled technicians, which can be costly and time-consuming. Both methods are invasive and/or lethal for the organisms under examination, and both suffer issues with subjectivity in taxon assignments. Despite the recent publication of standard methods (*Joy et al., 2013*; *New Zealand Ministry for the Environment, 2022*) there is still potential for inconsistency between sampling personnel and between laboratories. There is also a significant proportion of macroinvertebrate taxa which do not have assigned indicator values, leading to confusion and inconsistencies in analytical approaches, with some authorities excluding/omitting them from site calculations and some creating indicator values based on their professional judgment (*Stark & Maxted, 2007*). Importantly, both methods have limited taxonomic scope, and do not include bioindicators with rapid generation times that respond quickly to environmental disturbance, such as fungi, phytoplankton and bacteria which can be responsive indicators (*Kutty et al., 2022*). Despite the drawbacks, and through necessity, biological assessment of both fish and macroinvertebrate communities is standard and common practice, and both attributes are now mandatory monitoring requirements for regional and unitary councils under the NPSFM (*New Zealand Ministry for the Environment, 2020*).

To address the challenges and limitation of the MCI, we created a whole-ecosystem, stream health monitoring tool, and used the Chessman learning algorithm (*Chessman, 2003*) to train a taxon-independent community index (TICI) on an eDNA dataset from 53 well-studied riverine sites with 16 replicate samples per site and 8 metabarcoding assays per sample. We assigned indicator values for 3,000 commonly encountered amplicon sequence variants (ASVs) spanning a broad range of indicator groups, including bacteria, micro-eukaryotes, plants, fungi and animals. By comparing the TICI index values against historic MCI data, we show that the new index provides a precise and repeatable measure of ecological health that could be extended to other environments.

## MATERIALS & METHODS

### Sample collection and preservation

Environmental DNA samples were collected from 53 monitoring sites distributed throughout the North and South Islands of New Zealand between 14th December 2020 and 4th June 2021 (Fig. 1; *Melchior & Baker, 2023*). Of these sites, 44 were classified by sample collectors as 'hard-bottomed' (stony or boulders) and 9 were 'soft-bottomed',

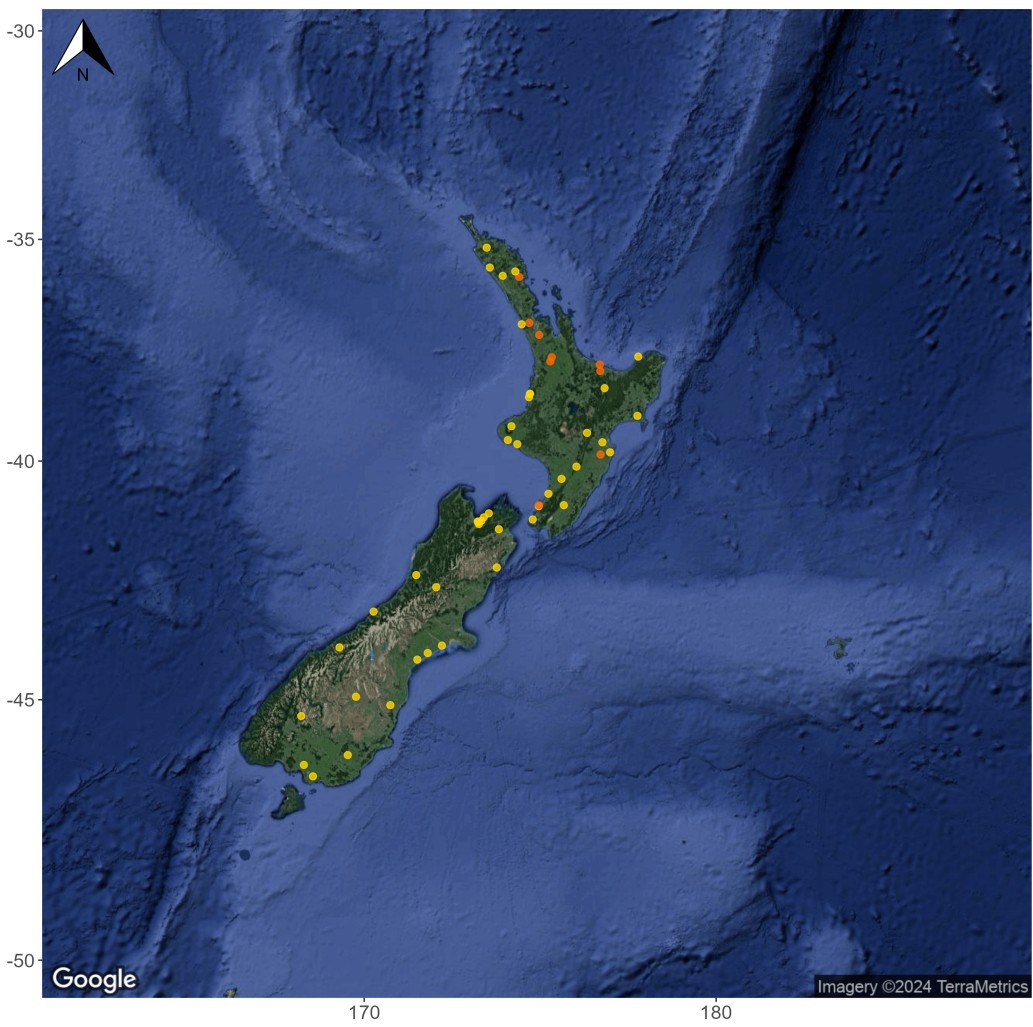

**Figure 1** **Geographic spread and habitat type of 53 monitoring sites included in the Summer 2020/2021 high-replicate eDNA survey.** Hard-bottomed streams are shown as yellow circles and soft-bottomed streams are shown as orange circles. 40 of the sampling locations were long-term Regional Council monitoring sites and had at least five years of historic MCI data available through the LAWA website (https://www.lawa.org.nz/). Map layer by Google (https://www.google.com) *via* the 'ggmap' R package (*Kahle & Wickham, 2013*).

with fine sediment substrate. Collections took place as near to base flow as possible, while avoiding sampling when water was visibly discolored from rainfall. Full site details including substrate types, coordinates, and sampling dates are provided in File S1. eDNA samples were collected using Wilderlab eDNA mini-kits, each containing an encapsulated 30 mm diameter, 1.2 μm cellulose acetate syringe filter with luer-lock inlet and outlet fittings, a 60 ml luer-lock syringe, 350 μl DNA/RNA Shield preservation buffer (Zymo) pre-loaded in a 3 ml luer lock syringe, and a pair of sterile nitrile gloves. 'Boosted' mini-kits contained two filters and two preservative syringes, but were otherwise identical to the standard mini-kits described above. The standard operating protocol for the Wilderlab mini-kit is available

in the accompanying Zenodo archive (doi: 10.5281/zenodo.7768777), and the protocol for the boosted kits follows the mini-kit SOP, except that two samples are taken and the lysate pooled together on receipt at the laboratory (note that this had no noticeable effect on detection sensitivity (*Melchior & Baker, 2023*), so boosted replicates are considered as standard replicates for the purposes of this study). For each sample, the 60 ml syringe was used to filter 1 L of water through the encapsulated filter (or a smaller volume if the filter clogged). Residual water in the filter was then purged by forcing 60 cc of air through the filter, using the emptied 60 ml syringe. The material collected on the filter was then preserved by injecting the 350 µl DNA/RNA Shield solution (Zymo) from the pre-loaded 3 ml syringe into the filter inlet. 16 eDNA samples were collected from each site: 8 using the standard method and 8 using the boosted method. Samples were sent to the Wilderlab sequencing laboratory in Wellington within three weeks of sample collection.

## DNA quality analysis and NGS library preparation

Sample lysates were extracted from the encapsulated filters by drawing the plunger back on the 3 ml syringe while holding the assembly vertically with the filter at the top to create a suction and waiting 5 s for the lysate to extract into the barrel of the 3 ml syringe. Sample lysates were then transferred into low-bind microcentrifuge tubes (Eppendorf) and stored at −20 °C prior to DNA extraction.

For DNA extraction and purification, 200 µl of each sample lysate were loaded into a Genolution GD141 cartridge and run on the Genolution Nextractor NX-48S system using the standard extraction settings. DNA quality/quantity analysis, adapter-fusion, indexing, and amplification were carried out in single-step quantitative PCR reactions on an Applied Biosystems QuantStudio 1 qPCR instrument, similar to the methods described in *David et al. (2021)* and *Urban et al. (2022)*. DNA extracts were PCR-amplified using eight fusion-tag mitochondrial and nuclear rRNA assays for the detection of vertebrate invert, plant, microeukaryote and microbial DNA (see Table 1 for primer sequences). Fusion tag primers included Illumina P5 and P7 adapter sequences, Illumina TruSeq™ sequencing primer bind site (forward primer only), unique 8 bp index sequences, and locus specific primers, respectively. PCR reactions were carried out in duplicate, with each reaction containing 5 µl SensiFAST 1 × LoRox SYBR Mix (Bioline), 0.25 µl forward primer (10 µM), 0.25 µl reverse primer (10 µM), 0.5 µl BSA (10 mg ml$^{-1}$, Sigma Aldrich), 2 µl deionised water and 2 µl template DNA. qPCR cycling conditions included an initial denaturation of 3 min at 95 °C, followed by 40 cycles of 5 s at 95 °C, 10 s at the annealing temperature specified in Table 1, and 15 s at 72 °C. DNA quality and quantity were confirmed by assessing that a sigmoidal log-amplification curve was visible at a Cq value of < 35. A negative control reaction containing 2 µl of deionised water in place of the template DNA was included with each sequencing run.

Sequencing libraries were pooled at approximately equimolar concentration using the final normalized ΔRn fluorescence values as a guide and cleaned and double-end size selected using AMPure XP magnetic beads (0.9 × and 1.2 × for lower and upper size bounds, respectively). The final pooled library concentration was determined using a Qubit 4 Fluorometer (ThermoFisher Scientific) and the concentration was adjusted to 50

Wilkinson et al. (2024), *PeerJ*, DOI 10.7717/peerj.16963

**Table 1** **Locus-specific sections of the fusion-tag metabarcoding primers used in this study.**

| Assay | Gene | Forward Primer | Reverse Primer | Anneal temp (°C) | Ref | No. TICI indicators | Proportion | Read depth (±SE) |
|-------|------|----------------|----------------|-------------------|-----|---------------------|------------|-------------------|
| BE | 18S-V9 | CCCTGCCHTTTGTACACAC | CCTTCYGCAGGTTCACCTAC | 52 | [1] | 682 | 22.7% | 6,687 ±153 |
| BU | 18S-V9 | TTGTACACACCGCCC | CCTTCYGCAGGTTCACCTAC | 52 | [1] | 719 | 24% | 6,177 ±200 |
| CI | COI | DACWGGWTGAACWGTWTAYCCHCC | GTTGTAATAAAATTAAYDGCYCCTARAATDGA | 45 | [2] | 796 | 26.5% | 7,621 ±184 |
| MZ | *rbc*L | CTTCTTCAGGTGGAACTCCAG | GTCACCACAAACAGAGACTAAAGCAAGT | 52 | [3] | 101 | 3.4% | 4,393 ±84 |
| RV | mt12S | TTAGATACCCCACTATGC | TAGAACAGGCTCCTCTAG | 58 | [4] | 35 | 1.2% | 4,444 ±127 |
| TP | *trn*L | GGGCAATCCTGAGCCAA | CCATTGAGTCTCTGCACCTATC | 52 | [5] | 173 | 5.8% | 7,343 ±170 |
| UM | 16S-V5 | GGATTAGATACCCTGGTA | CCGTCAATTCMTTTRAGTTT | 52 | [6] | 267 | 8.9% | 1,990 ±66 |
| WV | mt16S | GACGAGAAGACCCTWTGGAGC | CCRYGGTCGCCCCAAC | 58 | [7] | 227 | 7.6% | 11,377 ±201 |

**Notes.**

The composition of 3,000 TICI indicator ASVs by assay code is included in the two right-hand columns. Primer references are as follows: [1] *Amaral-Zettler et al. (2009)*; [2] Forward primer adapted from *Vamos, Elbrecht & Leese (2017)*; reverse primer this study; [3] *Bradley et al. (2007)*; [4] Forward primer from *Riaz et al. (2011)*; reverse primer from *Kelly et al. (2014)*; [5] *Taberlet et al. (2006)*; [6] Forward primer from *Morey et al. (2006)*; reverse primer from *Lane et al. (1985)*; [7] Forward and reverse primers adapted from *Nester et al. (2020)*.

pM in sterile DNAse/RNAse free water. The library was then loaded onto an iSeq i1 V2 reagent cartridge with a 300-cycle flow cell (Illumina) with 5% Phi X and run for 290 cycles in a single direction on an Illumina iSeq 100 instrument.

Sample extraction, PCR preparation and the PCR-cleanup-sequencing step were all carried out at separate, dedicated workstations, with the latter step occurring in a separate 'post-PCR' room. All benches were cleaned with freshly prepared 1:10 household hypochlorite bleach solution prior to use, and nitrile gloves were worn to avoid contamination.

## ASV generation and taxonomic assignment

The iSeq 100 output sequence fastq files were de-multiplexed in R (*R Core Team, 2021*) using the insect package (v 1.4.0; *Wilkinson et al., 2018*) and trimmed sequences were filtered to produce a table of exact amplicon sequence variants (ASVs) using the DADA2 package (*Callahan et al., 2016*) with chimeras removed using the 'consensus' method, the 'pool' parameter set to FALSE and a minimum sequence read count of 5.

For validation purposes, we calculated an eDNA-based MCI (eMCI) for each sample. This initially required us to identify each ASV to the lowest possible taxonomic rank. Taxon assignment followed a standard four-step classification process, involving: (1) exact matching against an intensively-curated database of previously detected ASVs; (2) remaining ASVs are exact-matched against a global reference sequence database primarily compiled of trimmed reference sequences downloaded from GenBank (*Benson et al., 2010*), BOLD (*Ratnasingham & Hebert, 2007*), and the RDP reference database (v18; *Cole et al., 2014*; accessed 15 June 2022 from https://doi.org/10.5281/zenodo.4310150; used for UM assay only), and matching ASVs assigned at the lowest common ancestor level (LCA; *i.e.,* assigned to genus level if matched with 100% identity to more than one species, or to family level if matched to more than one genus); (3) remaining ASVs >50 bp in length are matched with single indel/substitution tolerance against the same GenBank/BOLD reference database and matching ASVs assigned at LCA level; and (4) remaining ASVs are queried against the local GenBank/BOLD reference database using the SINTAX classification algorithm (*Edgar, 2016*) with a conservative assignment threshold of >0.99 and a maximum assignment resolution of genus level. Maximum genus-level assignment was enforced for the 18S and microbial 16S assays (assay codes BE, BU and UM; see Table 1) due to the low resolution/mutation rates and low coverage of reference sequences available for these markers. Immediately following the taxon assignment step, any DNA identified as having human origin was removed to alleviate any privacy concerns (raised in *Whitmore et al., 2023*).

## Selection of indicator ASVs and calculation of indicator values

The TICI uses a similar approach and rationale to the MCI, where the community of organisms at a site is assumed to reflect its ecological heath. In the case of MCI, each taxon has a predefined indicator value, and the index is essentially the average indicator value of the taxa found at the site (*Stark & Maxted, 2007*). Sites featuring communities dominated by 'hardy' taxa like worms and chironomids score poorly (since both *Chironomus* and

Oligochaeta have indicator values of 1.0), while sites dominated by pollution-sensitive taxa such as stoneflies, mayflies and caddisflies ('EPT' taxa) are assigned high scores, since many EPT genera have indicator values between 8 and 10. Indicator values are assigned objectively to each taxon in advance using an iterative learning approach known as the 'Chessman process', which incorporates known habitat quality information and the presence or abundance of the taxon across a range of sites (*Chessman, 2003*). For the taxon-independent 'TICI' approach, indicator values are assigned using the same *Chessman (2003)* method, except that these values are assigned to a large number and diversity of amplicon sequence variants (ASVs). These are distributed across the tree-of-life rather than being constrained to a comparatively small number of invertebrate taxa. Low scoring ASVs are associated with impacted sites, while high scoring ASVs assigned high scores are associated with pristine sites.

The 3,000 most commonly encountered ASVs (by presence/absence, and independent of taxonomic origin) were selected and assigned indicator values using the Chessman process (*Chessman, 2003*). The value of 3,000 was chosen based on a gradient optimisation approach, using different numbers of indicator ASVs to train TICI models. We selected two optimisation criteria to simultaneously maximise both accuracy and precision: (1) the correlation coefficient between the TICI and the 5-year median MCI (accuracy); and (2) the average coefficient of variation (CV) of replicate TICI values (precision). See File S2 for further details on the optimisation of indicator ASV numbers. The assignment of indicator values to ASVs means that a single taxon can be represented by multiple ASVs in cases where intraspecific sequence variation exists within a population. However, the large number of ASVs spread across the tree-of-life, and the use of multiple metabarcode markers, minimizes the impact of individual taxa on the final TICI value.

In our application of the *Chessman (2003)* process, we randomized the 53 sites to simulate a naive initial site condition ranking. We then applied following iterative process until convergence (*i.e.,* no change in indicator values between iterations):

1. For each of the 3,000 ASVs:
   (a) Rank the sites based on the number of replicate samples that the ASV appears in (ie an 'ASV prevalence' score ranging from 0–16)
   (b) Calculate the Spearman correlation coefficient between the site condition ranking and the ASV prevalence ranking, giving the indicator value for the sequence on a scale of −1 to 1 (where a score of 1 is a perfect correlation between ASV rank abundance and stream condition, a score of −1 is a perfect inverse correlation, and a score of 0 shows no correlation)
   (c) Convert indicator values to a 0–10 scale by adding 1 and multiplying by 5
2. Calculate the TICI for each sample, as the average indicator value of the ASVs found within the sample
3. Calculate the TICI for each site, as the average TICI of the samples taken from the site
4. Update the site condition ranking based on the site-averaged TICI values (from step 3) in ascending order.

The final TICI is then calculated for each sample by multiplying the mean indicator value of the indicator ASVs present in the sample by 20 to provide a stream condition

index on a scale consistent with the existing MCI. While this step is not strictly necessary, it helps with interpretation for existing MCI users.

Note that in this unsupervised learning approach (where the sites are randomised to simulate a naive initial site condition ranking) the final TICI values can end up being positively or negatively correlated with site condition, with equal probability. This is because the learning process can find the condition gradient in the data, but it can't determine the direction of the gradient (*i.e.,* which are the healthy sites and which are more impacted). An alternative site condition measure such as MCI can be used to determine if a high TICI score indicates good or poor site condition, and in the latter case, the indicator values can be inverted to achieve a positive relationship.

To test the effect of the initial site ranking on the final model parameters, we trained 10 different TICI models, each with the same 3,000 indicator ASVs, but with different randomised initial site-condition rankings. We also trained a model with the initial site condition ranking generated using an 'eMCI' (*i.e.,* MCI values derived using the eDNA data rather than by physical identification) to incorporate *a priori* condition information for the 53 sites used in the analysis. Note that the eMCI was used in place of the 5-year median kick-net MCI since only a subset of the sites had historic MCI data available, and the eMCI and MCI were closely correlated for those 40 monitored sites (see File S3 for eMCI methods and output). The TICI values from the 11 different models were then compared using pairwise linear regression (lm function in base R; *R Core Team, 2021*).

To validate the TICI index and compare it against the current standard stream condition assessment method, site-averaged TICI values were correlated against the 5-year median MCI for the subset of sites for which multiple-year MCI data were available (40 of the 53 sites), using linear regression as above. Site-averaged TICI values were used in place of individual sample TICIs for model fitting to avoid pseudo-replication issues.

As a quality control measure, we imposed a two-sigma rule, where if the number of TICI ASVs (*i.e.,* from the set of 3,000 indicator sequences) appearing in the negative control exceeded two standard deviations from the average across all runs, the DNA extractions, library prep and sequencing run would be repeated. This did not occur during this study, with all negative control reactions below the two-sigma threshold (note that some TICI ASVs originate from airborne microbes such as *Aspergillus,* so a base number of ASVs in the negative control is normal).

All R code and input data for analysis are included in the accompanying Zenodo archive (doi: 10.5281/zenodo.7768777).

## RESULTS

A total of 848 eDNA samples were collected and analyzed from the 53 sites, with 8 metabarcoding assays run on each sample and 42.2 million sequence reads passing the DADA2 quality filter (raw sequence data available in SRA archive at https://www.ncbi.nlm.nih.gov/sra/PRJNA950216). This constituted 762,900 count-aggregated sample-ASV records. Of the 3,000 most commonly encountered ASVs (in terms of presence/absence rather than total ASV counts), the taxon assignment process identified

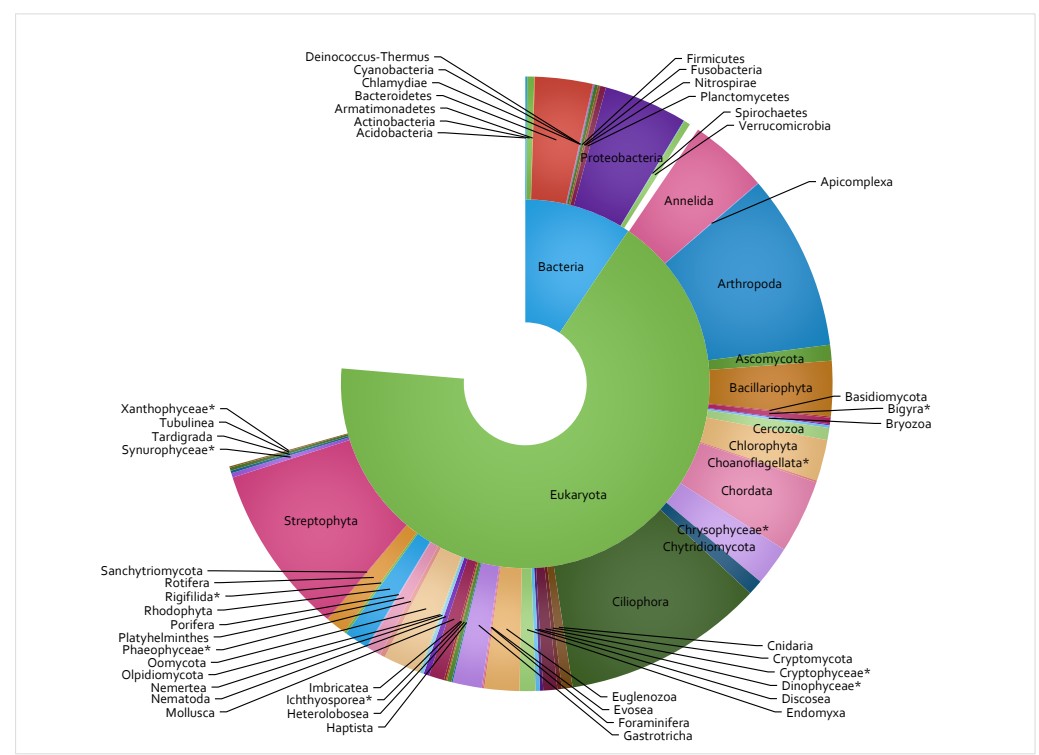

**Figure 2** Taxon IDs for 3,000 TICI indicator ASVs. The 3,000 most commonly encountered ASVs across the 8 metabarcoding assays are shown at superkingdom (inner ring) and phylum (outer ring) levels. Missing segments indicate the number of indicator ASVs that could not be identified at each taxonomic rank, and asterisks show lower-ranked taxa whose phyla are not specified in the NCBI taxonomy database.

**Table 2** Arbitrary stream condition categories to aid in the interpretation of TICI values.

| TICI range | Stream condition category | No. samples in category | No. sites in category |
|---|---|---|---|
| <80 | Very poor | 123 | 8 |
| 80–90 | Poor | 130 | 8 |
| 90–100 | Average | 183 | 11 |
| 100–110 | Good | 137 | 8 |
| 110–120 | Excellent | 175 | 12 |
| >120 | Pristine | 100 | 6 |

**Notes.**
Roughly similar numbers of samples and sites are in each of six categories ranging from very poor to pristine.

51 phyla (Fig. 2), 100 classes, 236 orders, 344 families, and 396 genera. The COI primer set (assay code CI) was the most well-represented assay, accounting for 796 (26.5%) of the 3,000 TICI indicator ASVs. This was followed by the two 18S assays: BU and BE, accounting for 24% and 23% respectively. The vertebrate-specific 12S RV ecoprimers (*Riaz et al., 2011*) were the most under-represented, accounting for only 1.2% of the indicator ASV set (35 indicator ASVs; Table 2).

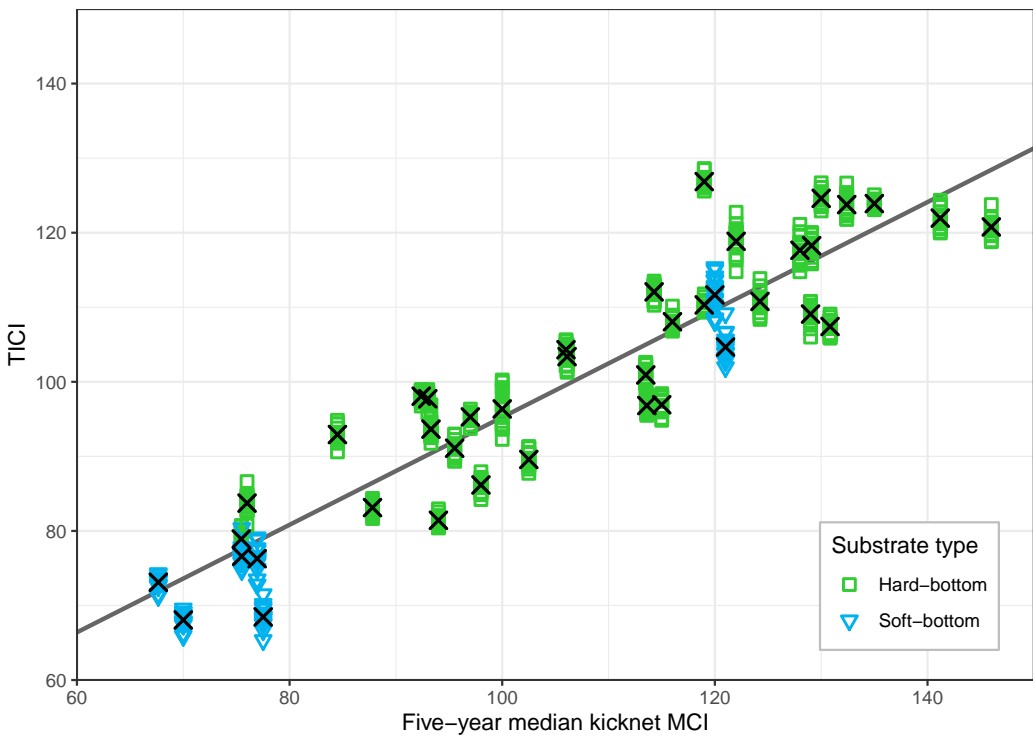

**Figure 3** **Validation of the TICI index against existing five-year median kick-net MCI values for the 40 sites where historic MCI data were available.** TICI values from 640 individual samples (40 sites × 16 replicate samples) are plotted against their kick-net MCI site-medians. The linear regression of five-year median kick-net MCI *versus* site-averaged TICI values (shown here as black crosses) was significantly correlated with a *p*-value of less than 0.001 and an adjusted $R^2$ value of 0.825.

There was a strong correlation between the site-averaged TICI values and the five-year median MCIs, with an adjusted $R^2$ value of 0.825 ($p < 0.001$; Fig. 3). This trend appeared to be independent of substrate type, noting that only 9 of the 53 sites featured soft-bottomed habitat. There was minimal variation between the TICI values of replicate samples, with a mean coefficient of variation of 0.0125 (see File S1). The indicator scores of the TICI ASVs found in each sample were approximately beta distributed, with scores skewed towards zero in highly impacted sites and skewed towards ten in reference sites (Fig. 4). Site-averaged TICI values ranged from 68.05 for the Papanui Stream in the Hawke's Bay, to a maximum of 126.86 for the Haast River on the West Coast of the South Island.

To provide users with an intuitive and accessible stream condition ranking system, we set six categories based on the range of TICI values among the 53 sites and 848 samples, with round-numbered thresholds and roughly similar numbers of samples in each. The ranking system ranges from very poor to poor, average, good, excellent and pristine. The categories and TICI thresholds are shown in Table 2.

The number of TICI ASVs present per sample varied by site, with an overall average of 373 and a standard deviation of 135. There was a positive relationship between the number of TICI ASVs per sample and the precision of the TICI values around the site mean,

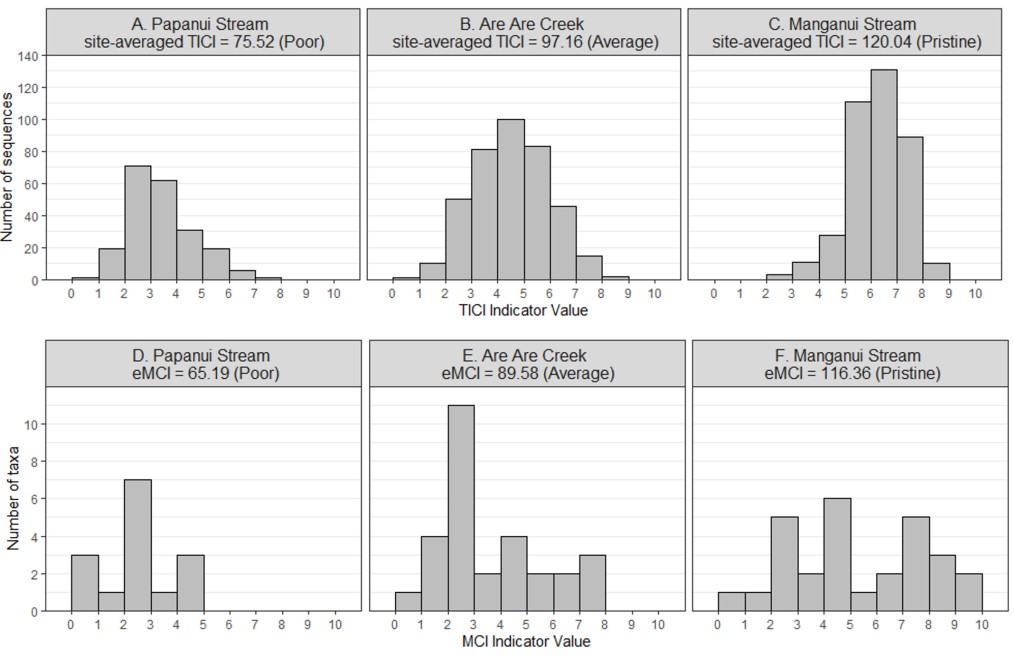

**Figure 4** **Example distributions of TICI and eMCI indicator values.** The TICI indicator values (A–C) follow a smoother beta distribution than the eMCI (D–F) indicator values due to the comparatively larger size of the indicator set. Shown here are representative eDNA samples taken from poor (A, D), average (B, E) and pristine (C, F) sites. Example sites shown in the figure are the Papanui Stream in the Hawke's Bay (site-averaged TICI = 75.52), Are Are Creek in Marlborough (97.16), and Manganui Stream in the Southern Waikato Region (120.04), respectively.

with samples containing more than 100 TICI ASVs providing tighter clusters (Fig. 5). For samples containing 100 or more unique indicator ASVs, 99.6% of samples (843/848) produced TICI scores within four units of the 'true value' (*i.e.,* the site-averaged value over 16 sample replicates), while 97% of samples (821/848) were precise to within three units. For samples containing fewer than 100 unique indicator ASVs 94% (15/16) and 87.5% (14/16) were precise to within four and three units, respectively.

Of the ten Chessman process runs where the initial condition rankings of the sites were randomised, all converged on a solution where the 3,000 indicator values were either identical (two of the 10 permutations) or almost identical ($R^2 = 0.99$; 8 of 10) to the eMCI-guided TICI model (where the sites were initially ranked on stream condition using the site-averaged environmental MCI values).

While taxon assignment is not necessary for calculating the TICI index, it is useful to observe the taxonomic origin of the ASVs with significantly high or low indicator values. The ASVs that were associated with poor stream condition were dominated by oligochaete worms, including *Bothrioneurum vejdovskyanum* (indicator value = 1.08), *Potamothrix bavaricus* (1.42) and *Ilyodrilus* sp. (1.99). Other low scoring taxa included the Australasian swamphen or pūkeko *Porphyrio melanotus* (1.69), the shortfin eel *Anguilla australis* (1.70) and the copepod *Acanthocyclops robustus* (1.75). High scoring taxa included

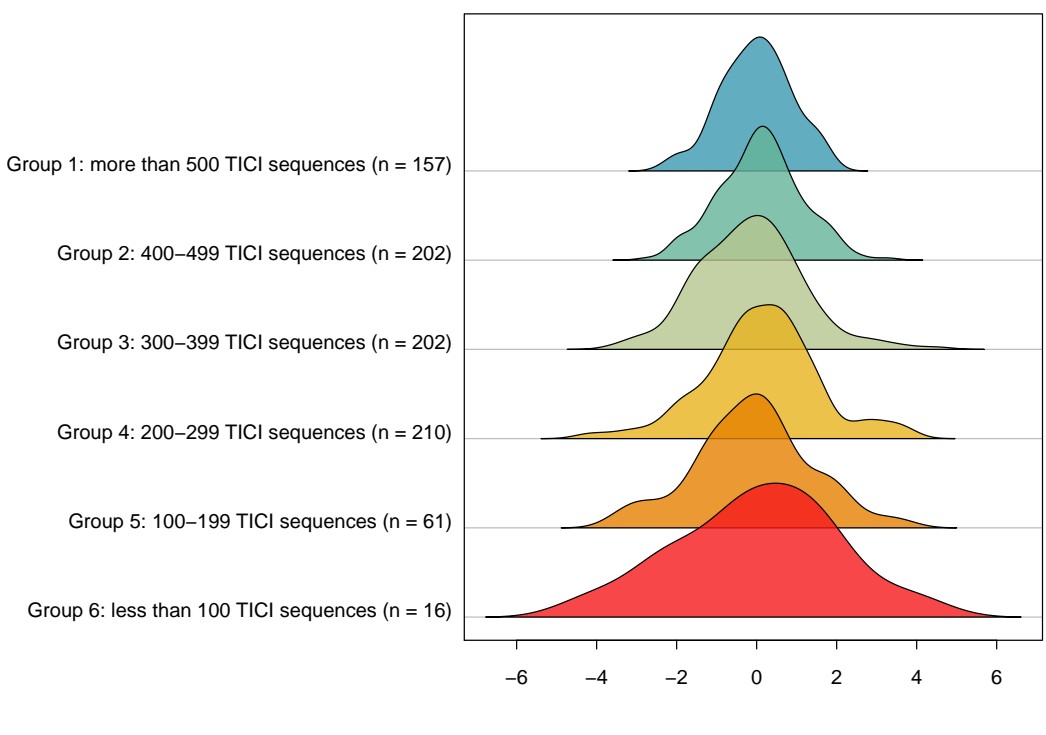

**Figure 5 Ridgeline plot showing a positive relationship between the number of indicator ASVs per sample and the precision of the site-averaged TICI.** A total of 99.7% of samples containing 100 or more indicator ASVs yielded TICI values within four units of the site-average (average over all 16 site replicates), while 94% of samples containing fewer than 100 TICI ASVs were within a similar level of precision. Plot produced using the 'ridgeline' R package (*Soage González & Koncevicius, 2023*).

several 'EPT' taxa (mayflies, stoneflies and caddisflies; orders Ephemeroptera, Plecoptera and Trichoptera, respectively), including *Nesameletus* sp. (8.91), *Psilochorema macroharpax* (8.03) and *Ameletopsis perscitus* (7.92). Several ASVs of probable microbial origin were also included in the taxa with high indicator values, but most of these could not be resolved to a low taxonomic rank. These included an ASV classified to the phylum Bacteroidetes (indicator value = 8.3), one to class Betaproteobacteria (8.02), and one to order Enterobacterales (8.12). The high scoring ASVs also featured a number of ciliates that could not be resolved to species level, including one resolved to the class Spirotrichea (8.28) and another to the order Pleurostomatida (class: Litostomatea; indicator value = 7.65; see File S4 for a full list of the 3,000 ASVs with their TICI indicator values and assigned taxon IDs).

## DISCUSSION

eDNA metabarcoding offers extraordinary utility for surveying across the tree-of-life. Metabarcoding assays can be tailored to target specific taxa or taxon groups with excellent sensitivity and resolution. However, long lists of unfamiliar taxa can be difficult to interpret and translate into management practice. eDNA-based indices can alleviate these issues, but

often do not make full use of eDNA data, a large fraction of which cannot be identified with current underpopulated reference databases. To address this, we applied a taxon-independent approach to multi-assay eDNA metabarcoding data, assigning indicator values to amplicon sequence variants (ASVs) rather than to taxa. The resulting taxon-independent community index (TICI) is a precise, accurate and cost-effective ecological condition indicator that makes thorough use of both assignable and unassignable eDNA metabarcoding output.

Biotic indices are designed to reduce complex, multidimensional data to a single value. Their interpretation in ecosystem management requires an understanding of which variable or suite of variables the index is measuring, which can then inform the appropriate management action (*Pawlowski et al., 2018*). For example, the MCI index in widespread use for stream health assessment in NZ, is described as a measure of organic enrichment (*Stark, 1985*); although it is acknowledged that the index will respond to other stressors such as sedimentation, low flow, alterations to instream or riparian habitat—indeed any of the multitude of stressors associated with land use change (*Stark & Maxted, 2007*). The MCI may also respond to stressors associated with presence of invasive species such as the diatom *Didymosphenia* (Didymo), which has been shown to affect invertebrate community composition (*Kilroy, Larned & Biggs, 2009*). For environmental management purposes, it is therefore useful to define ecological health broadly as an overarching measure encompassing a suite of inter-related biotic and abiotic variables, including organic enrichment, inorganic pollutants, oxygenation, nativeness, and the presence/absence of opportunistic or invasive species. In our study, the Chessman process extracted the dominant gradient from the eDNA data without any *a priori* knowledge of site condition. We suggest that this gradient is a composite of the different ecological and environmental factors that contribute to stream condition, and can be broadly defined as the 'ecological health' of the system.

From a management utility perspective, the combined eDNA output including both taxon-independent and taxon-dependent results provide a useful triaging tool. We envisage that an anomalous TICI score could be interrogated further using the full list of taxa detected, in concert with other tools such as nutrient/oxygen analysis and fecal source tracking indicators (FST; *e.g.*, *Gomi et al., 2014*; *Moriarty et al., 2011*). For example, inspection of the taxonomic eDNA results from a lowland stream with a declining TICI value and abnormally high turbidity, could reveal the presence of the invasive koi carp *Cyprinus rubrofuscus*. This species is known to stir up sediment while feeding, driving changes in the macrophyte community (*Koehn, Brumley & Gehrke, 2000*). In this case, the magnitude of the decline in TICI could be used as a first step in a decision-making process. A mild change might be an appropriate trigger for gathering complementary biotic and abiotic data from the site, whereas larger TICI change could be an efficient way to trigger immediate (and early) intervention by resource managers.

A major challenge facing the eDNA field is how to make the technology accessible to decision makers and environmental stakeholders. We advocate that the TICI should be used in a similar way to the morphology-based MCI before it: to provide scalable and accessible data on states and trends at a macro-environmental level. In our experience working with community conservation groups, the most commonly-asked questions typically involve

the presence or absence of a few culturally-important species, the current state of the waterway, and whether it is improving or deteriorating. Used in conjunction with typical taxonomic eDNA outputs, the TICI can be used as a reliable indicator to help address these basic questions. For example, in a recent study, the TICI and associated taxonomic eDNA outputs were used to assess the progress of a community-led riverine restoration project toward improving stream health, with a special focus on threatened species such as the 'Whio' blue duck (*Hymenolaimus malacorhynchos*), and reveal the impact of Tropical Cyclone Gabrielle on the condition of the waterway (*Drysdale et al., 2023*). We believe that the next challenge is to interrogate the increasingly large tree-of-life datasets being generated by national-scale eDNA monitoring efforts, to develop a suite of other indices that address a broader range of questions and environmental challenges. These could include changes in biotic state (*e.g.*, new taxa or altered taxon-association networks), an exploration of native *versus* introduced species, changes in relative abundance, and seasonal changes in species composition (*e.g.*, *Sander et al., 2023*; note that after running several thousand additional samples, the TICI appears to remain highly stable throughout the year; see wilderlab.co.nz/explore to view publicly available samples with associated TICI values). Our limited sample size of 40 intensively-monitored sites (*i.e.*, those that had more than 5 years of historic MCI data available) precluded us from exploring these patterns further, but the introduction of the TICI in 2022 helped to stimulate a much larger investment from the NZ Ministry for the Environment, involving the collection of a further 3,000 eDNA samples from New Zealand's network of riverine monitoring sites (see https://wilderlab.co.nz/explore/). We anticipate that the availability of these data will enable further refinement of indices that provide extra context to the macro-environmental patterns revealed by the TICI.

The TICI index offers high precision as demonstrated by low variance in scores between replicate samples. This feature was particularly evident in the 98% of samples for which at least 100 unique indicator ASVs were detected. The distribution of indicator scores within samples were distributed as non-skewed normal distributions (see Fig. 4), with the TICI index derivable from the mean parameter without bias. A useful feature of this precision is that it can make use of samples collected with lower levels of replication than needed in taxon-dependent eDNA metabarcoding studies (*Zinger et al., 2019*). We generally advocate for six eDNA replicates collected in the Summer months as a standardized unit of sampling effort, based on a previous species accumulation analysis of this dataset (*Melchior & Baker, 2023*). Adequately replicated samples provide more precise results (*i.e.,* by averaging TICI scores over six replicates), enable assessment of inter-replicate variation, and provide additional confidence to detect taxon-specific target and indicator species. However, in practice, different organizations use eDNA for a variety of reasons, often lowering the sample replication number to maximize spatial coverage or work within limited budgets.

Low replication in eDNA biomonitoring is particularly common for surveys conducted by community-based conservation groups, who may lack access to sufficient funding to follow best-practice guidelines, but whose motivations are more focused on engaging, educating and promoting environmental awareness (*Mascia et al., 2003*). The inclusion of high-precision, low-replication samples in a national reporting framework could greatly

enhance our national state and trend monitoring of stream condition. Further bridging the divide between community science and professional monitoring comes with several additional benefits, including fostering beneficial collaboration between government agencies and volunteer groups, fulfilling policy requirements, and improving attitudes through helping people engage and connect with their environment (*Dickinson et al., 2012*).

The cost of high-throughput sequencing has continued to subside in recent years, predominantly driven by the human genomics field (*Reuter, Spacek & Snyder, 2015*). Environmental genomics have benefited from this trend, with the rapid global uptake of eDNA monitoring and new eDNA studies (*Takahashi et al., 2023*). At the time of writing, the cost to a NZ agency for running the 8 TICI metabarcoding assays on six replicate samples is approximately $1500 NZD ($950 USD; *i.e.,* one site at one annual Summer monitoring time-point; see https://wilderlab.co.nz/order). This compares favorably to the costs of deploying insect and fish survey teams and analyzing a single bulk insect sample for MCI, while also providing a slew of other information about aquatic and terrestrial species in the catchment. The cross-usage of resources between sectors and interagency teams also presents a unique opportunity to reduce duplication, dramatically improve cost efficiency (*Lindenmayer et al., 2014*) and assist in defragmenting environmental reporting (*New Zealand Parliamentary Commissioner for the Environment, 2019*). Multiple needs can be addressed at once including state-of-environment trend assessment, biosecurity surveillance, threatened species monitoring, and education and awareness. For example, a single round of sampling carried out by an agency's fish monitoring team can help to fulfill the needs of the same agency's biosecurity team for pest surveillance, as well as provide useful distribution data for another agency's threatened species monitoring, and even provide a nearby school with valuable information on the presence of culturally important food sources and other species of interest in their local stream. Improving the accessibility of eDNA tools and data can even facilitate new scientific discoveries, such as the effects of deforestation on arthropod communities (*Waters, Ni & McCulloch, 2023*).

A common objection to eDNA metabarcoding for compliance and regulatory monitoring is that false positives and negatives lead to misleading results (*Darling, 2020*). This is especially prevalent in biosecurity applications where a false alarm can trigger an expensive incursion response. eDNA can offer significantly lower false negative rates than traditional monitoring methods (*David et al., 2021*; *Jerde et al., 2011*), but eDNA signals can occasionally appear from non-living sources such as fecal matter from mobile predators, food, discharges of contaminated water, and sample mishandling (*Jerde et al., 2011*). Most environmental decision makers work in frameworks that tend towards accepting false negatives over false positives. Expectations tend to be set higher for eDNA than traditional monitoring methods, due to the 'burden of proof' faced by a new and disruptive technology (*Darling, 2020*). The taxon-independent approach outlined here addresses both the false positive and false negative objections. The large pool of possible indicator ASVs provides a great deal of redundancy in generating the TICI scores. This means that contaminant ASVs present at low copy numbers cannot significantly shift the index value, and even if several important indicators are missed, the index is still informed by alternative ASVs.

In the current study, our negative control reactions contained some airborne yeasts such as *Aspergillus*, microbial DNA and vascular plant DNA (primarily *Pinus*, which is highly abundant and pollenates heavily in New Zealand; control data included in SRA archive https://www.ncbi.nlm.nih.gov/sra/PRJNA950216). Some of these ASVs appeared in the final 3,000-sequence TICI indicator set (for example two ASVs identified as *Pinus* are in the final set with indicator values of 5.56 and 5.57 for the TP and MZ assays, respectively). With the exception of human DNA (which is removed during the taxon assignment step), we made the decision to leave ASVs that appeared in negative controls in the indicator set. This is to minimize subjectivity in indicator selection, and because we believe that in real samples with a background of high-concentration eDNA, airborne contaminants will have minimal bearing on the index.

Robust environmental monitoring and decision making requires a rigorous and reproducible method that is free from subjective judgment (*McDonald-Madden, Baxter & Possingham, 2008*; *Lindenmayer, Likens & Franklin, 2010*). Current morphology-based monitoring methods are dependent on expert judgment for taxonomic identification, habitat classification, and even indicator value assignment. For example, the current fish-IBI and MCI protocols in use for freshwater bioassessment in New Zealand require staff with extensive and consistent taxonomic training, and the regular publication of standardized methods designed to minimize operator variability (*Joy et al., 2013*; *New Zealand Ministry for the Environment, 2022*). Despite this, identification keys for many invertebrates are unavailable, even at high taxonomic ranks (*e.g.*, family, order and class). Furthermore, before the release of the *New Zealand Ministry for the Environment (2022)* guidelines, there was no expectation that identifications were confirmed. Some of the MCI indicator values are currently based solely on expert opinion, making reproducibility especially difficult. An additional issue with current MCI methodology is the often-subjective distinction that must be made between hard and soft-bottomed habitat types, each of which invokes a differing set of taxa and indicator values. Perhaps the most problematic feature of the status quo is that no replication is carried out for current MCI sampling protocols, and virtually all stream condition assessments feeding into the national state-of-environment reporting system are based on singleton samples that typically require a 5-year sliding window to detect change. In contrast, all of the 3,000 TICI indicator values in this study were assigned objectively, and eDNA sample replication is a standard feature of virtually all existing local and central government monitoring programmes (with six replicates having emerged as the industry standard; *Melchior & Baker, 2023*).

Can TICI-like approaches be extended into other applications and environment types? In the current study, the presence of multi-year condition data to train the index against was a key starting point. The MCI index has been a staple for stream health monitoring in New Zealand for several decades (*Stark & Maxted, 2007*), and so provided an ideal reference point for the initial site quality ranking step and the post-training validation. Through a global lens, most aquatic environment types in most regions around the world have no ecological health monitoring in place—this makes the extension and validation of TICI-like methods challenging. A notable exception is in Europe, where the DNAqua-Net initiative has made major inroads into eDNA method development and

uptake (see https://dnaqua.net/ for information and publications). Fortunately, we found that sites do not need to be initially ranked using an existing condition metric to train a TICI index, meaning that robust baseline condition data may not always be necessary to successfully refine and train a TICI-like model in a new environment. We envisage that qualitative assessment of condition, followed by multiple iterations to select TICI ASV's might provide a good starting point for the development of an ecological health index. Presenting a crude index may be better than having no data at all, and importantly, such an index may enable triaging and prioritization of aquatic environments for further study. Ultimately, validation of indexes will be needed to tie TICI-like approaches to ecological condition—this provides an opportunity for molecular, morphological and ecological disciplines to work more cohesively. Rapid advancements in machine-learning AI-tools may further assist in interweaving technologies and is well suited to data-rich application like eDNA, where most practitioners focus on a small fraction of the biodiversity that can be discovered using this technology. A TICI-like approach may also provide a powerful tool when measuring ecological integrity and trajectory—these are core attributes being explored for the technological back-end needed to support effective biodiversity credits and accreditation schemes.

## CONCLUSIONS

We set out to train a taxon-independent community index (TICI) that uses both identifiable and unidentifiable eDNA sequence data to produce an accurate, precise and cost-effective ecological health score. Our TICI index correlates well with the existing stream condition scoring method (MCI), while producing minimal inter-replicate variation. The robustness of the measure means that eDNA samples collected for a range of purposes by different organisations can be used as a novel way to draw a national picture of environmental state and trend. The TICI was tested and proven for ecological health assessment in New Zealand rivers, and future work could train similar models for any environment where eDNA can be collected and sites of different ecological condition can be identified to train the model. Further development of TICI-like methodologies offer a powerful pathway for the integrated, whole-ecosystem monitoring required to set explicit and measurable goals to reverse global trends in global biodiversity loss.

## ACKNOWLEDGEMENTS

We thank the Regional Authorities who contributed in-kind resources towards this study, including Auckland Council, Bay of Plenty Regional Council, Environment Canterbury, Environment Southland, Greater Wellington Regional Council, Hawke's Bay Regional Council, Horizons Regional Council, Marlborough District Council, Nelson City Council, Northland Regional Council, Otago Regional Council, Taranaki Regional Council, Waikato Regional Council and West Coast Regional Council. We also thank the field teams who collected the eDNA samples, including Alex Going, Alice West, Andrew Horrell, Ange Chaffe, Ariana Mackay, Ashley Mitchell, Alton Perrie, Asita Langi, Caitlin Reid, Caitlin Wheeler, Callum MacKenzie, Cassie Newman, Chris Drake, Ciaran Campbell, Dan

Montgomery, Darien Kissick, Erin Bocker, Ebi Hussain, Elise Reid, Emma Reeves, Fleur Tiernan, Ian Hurst, Jarred Arthur, John Prince, Jonny Horrox, Keeli MacKereth, Kelly Le Quesne, Lauren Visser, Maddison Jones, Noah Davis, Paddy Deegan, Paul Fisher, Peter Hamill, Scott Butcher, Stefan Beaumont and Warrick Johnston.

### Funding

This work was supported by a Callaghan Innovation Career Grant, awarded to Wilderlab NZ Ltd. for Amy A Gault (Grant no. WNZAI2002). The funders had no role in study design, data collection and analysis, decision to publish, or preparation of the manuscript.

### Grant Disclosures

The following grant information was disclosed by the authors:
Callaghan Innovation Career Grant, awarded to Wilderlab NZ Ltd. for Amy A Gault: WNZAI2002.

### Competing Interests

Shaun P. Wilkinson, Amy A. Gault and Susan A. Welsh are current employees of Wilderlab NZ Ltd., a commercial eDNA processing laboratory. Megan Shaffer was employed by Wilderlab NZ Ltd. during the course of this study. Joshua P. Smith is an employee of Waikato Regional Council, Hamilton, New Zealand. Bruno O. David was employed by Waikato Regional Council, Hamilton, New Zealand, during the course of this study. Andy S. Hicks is currently employed by the Ministry for the Environment, Wellington, New Zealand, and was employed by Hawke's Bay Regional Council, Napier, New Zealand, during the course of this study. Daniel R. Fake was employed by Hawke's Bay Regional Council, Napier, New Zealand, during the course of this study. Alastair M Suren is employed by Bay of Plenty Regional Council, Whakatāne, New Zealand.

### Author Contributions

- Shaun P. Wilkinson conceived and designed the experiments, performed the experiments, analyzed the data, prepared figures and/or tables, authored or reviewed drafts of the article, and approved the final draft.
- Amy A. Gault conceived and designed the experiments, performed the experiments, analyzed the data, prepared figures and/or tables, authored or reviewed drafts of the article, and approved the final draft.
- Susan A. Welsh analyzed the data, prepared figures and/or tables, authored or reviewed drafts of the article, and approved the final draft.
- Joshua P. Smith conceived and designed the experiments, performed the experiments, authored or reviewed drafts of the article, and approved the final draft.
- Bruno O. David conceived and designed the experiments, performed the experiments, authored or reviewed drafts of the article, and approved the final draft.
- Andy S. Hicks conceived and designed the experiments, performed the experiments, authored or reviewed drafts of the article, and approved the final draft.

- Daniel R. Fake conceived and designed the experiments, performed the experiments, authored or reviewed drafts of the article, and approved the final draft.
- Alastair M. Suren conceived and designed the experiments, performed the experiments, analyzed the data, authored or reviewed drafts of the article, and approved the final draft.
- Megan R. Shaffer conceived and designed the experiments, performed the experiments, authored or reviewed drafts of the article, and approved the final draft.
- Simon N. Jarman analyzed the data, authored or reviewed drafts of the article, and approved the final draft.
- Michael Bunce conceived and designed the experiments, analyzed the data, authored or reviewed drafts of the article, and approved the final draft.

## DNA Deposition

The following information was supplied regarding the deposition of DNA sequences:

The Amplicon Sequence Variants (ASVs) are available at NCBI BioProject: PRJNA950216.

## Data Availability

The R code and input files are available at Zenodo: Shaun P. Wilkinson, Amy A. Gault, Susan Welsh, Josh S. Smith, Bruno O. David, Andy S. Hicks, Daniel R. Fake, Alastair M. Suren, Megan R. Shaffer, Simon N. Jarman, & Michael Bunce. (2023). TICI: a taxon-independent community index for eDNA-based ecological health assessment. https://doi.org/10.5281/zenodo.10396304.

## Supplemental Information

Supplemental information for this article can be found online at http://dx.doi.org/10.7717/peerj.16963#supplemental-information.

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
