# Peer review of "TICI: a taxon-independent community index for eDNA-based ecological health assessment"

_PeerJ, doi:10.7717/peerj.16963_

## Round 0.1 · original submission · Major Revisions

Two recognized experts have assessed your manuscript and identified a number of issues that make the manuscript unacceptable in its present form. The most important aspects are that methods are not described in sufficient detail, the assessment of the performance of TICI through correlations with MCI, and that more information is needed to follow your conclusions drawn and to highlight also limitations of the study.

On the other hand, the reviewers underlined the unique position and great importance of your study, so I hope that their criticisms will allow you to carry out a substantial revision of the manuscript, which is a precondition for acceptance of the manuscript.

Reviewer 1 ·

Basic reporting

Wilkinson et al. present an interesting, important and timely paper on taxonomy-free eDNA-based stream monitoring. The approach and findings are presented in professional English. However, the reporting of Materials and Methods is not sufficiently clear (more details below). I consider hypotheses, results, literature references as appropriate.

I could not find access details to raw data. Beside raw data, all datasets necessary for the repetition of the study should be made publicly available, for example, the association of sampling locations with the 3000 ASV sequences used as TICI indicators.

Experimental design

I consider the research of Wilkinson et al. as original and within the aims and scope of PeerJ. I find the research topic well defined, relevant & meaningful. The authors state how their research fills an identified knowledge gap.

The research was done along high technical standards.

However, the methods are not described in sufficient details. I still struggle to understand how TICI is actually calibrated. One needs to know something about a site to calibrate a quality index, but from the MS it is not clear to me how this was done. Reading the supplementary materials helped with this, but references to relevant parts are not well described (e.g. the description of the eMCI index in Supplementary file S3). This is rather easily solved: more details need to be provided about the rationale, and the actual how-to of the methods. I have no doubt that dealing with this issue will be a simple task. In addition to providing a better description of the methods, I request the authors to provide an overview graph of what and why was done. This will benefit the readers a lot.


The few specific comments below will demonstrate my confusion:
297-298: Optimization criteria: what was optimized?

L295: what is the rationale behind the Chessman iteration process? Summarise in 2-3 sentences, so the reader does not need to read the cited paper.

296-299: What is the rationale behind correlating indicator ASV numbers with MCI, or with the coefficients of variation? What is optimized here and why should it be done this way?

L298 seems to be about conventional, morphological identification-based MCI values (as the medians of biomonitoring results from 5 years), but the next sentence (L301-305) seems to be talking about MCI values obtained fro ASVs.

L317-327: are these steps the actual Chessman iteration process? What is iterated here exactly? A script would help a lot with understanding of what happens here.

L319: prevalence score: why between 0-16? Is it because 8 samples were collected per site using the standard method and 8 using the boosted method? What is the difference between the efficiency of standard and boosted? Is boosted recording more taxa? Why can you handle these two sample types together?

L295: assigned indicator value: how? Related to my confusion regarding the Chessman iteration process: is the iteration assigning indicator value to the taxa, stream condition to the sites?

L320: site condition ranking: It is not clear to me how the site conditions were calculated. Did you assume site quality? Did you calculate this from eMCI?

In general: I (and maybe the regular reader) would benefit from a description of MCI and eMCI in the Methods. Information on this is currently spread around the MS and supplement, and for an MCI-naive reader it is difficult to make sense of it. Make sure that you use the same nomenclature throughout, e.g. is molecular MCI from L273 the same as eMCI?

L326-7: the previous steps of the process focus on ASVs, but the last two lines seem to be concerned with the site. How are the ASV-specific steps related to the site-specific steps?

L326: Calculate new site-TICI values: what were the old TICI values?

532-536: Again, I cannot follow: when site conditions were randomized, what did you use to train the TICI?

Validity of the findings

Why can you assess the performance of TICI through correlations with MCI, if the number of ASVs used for calculating TICI was determined by correlations with MCI? Isn't that circular? Am I missing something here?

L388: While taxon assignment is not necessary... taxonomy-independence is a key message of the paper, but after reading the methods this confuses me: it seems that TICI calibration relies on MCI which seems to be inherently taxonomy-derived. I think this is related to my confusion about the methods description above. Is it the case that TICI actually needs only a ranking of sites based on quality (or any other continuous criterion), and it just happens that this ranking here is done with MCI? If yes, it should be made clear for the reader that any site ranking will work just as well, for example, heavy metal loads, pH, agriculture intensity in the catchment, dissolved organic matter, etc. This will also help the authors to generalize about the use of their methods at sites were baseline condition data are not available (see L533).

L527-536: Develop this paragraph better - I consider it as a key part of the discussion. The main scope of such a taxonomy-free eDNA biomonitoring approach is application in high biodiversity areas where one does not have baseline environmental conditions, or sufficient taxonomic knowledge. How should we go about this?

·

Basic reporting

The MS "TICI: a taxon-independent community index for eDNA-based ecological health assessment” is well structured and written in clear and professional English. Tables and figures are mostly very good (but see on improved terminology of reads vs. sequences vs. ASVs in some of these below, section 4). The article's content can be assessed with the information provided. I could access the SRA data and R scripts. What is missing for the supplement are per-assay read numbers (mean / replicates...) and negative control handling + reference data bases (see section 3).

Regarding literature references / sufficient field background / context provided: I would advise to briefly look at several of the works published 3-5 years back in Europe on this subject (index development). Especially Pawlowski et al. 2018 provide an in-depth view on exactly this topic. I list three key citations below:

Hering, D., Borja, A., Jones, J. I., Pont, D., Boets, P., Bouchez, A., Bruce, K., Drakare, S., Hanfling, B., Kahlert, M., Leese, F., Meissner, K., Mergen, P., Reyjol, Y., Segurado, P., Vogler, A., & Kelly, M. (2018). Implementation options for DNA-based identification into ecological status assessment under the European Water Framework Directive. Water Res, 138, 192–205. https://doi.org/10.1016/j.watres.2018.03.003

Pawlowski, J., Kelly-Quinn, M., Altermatt, F., Apotheloz-Perret-Gentil, L., Beja, P., Boggero, A., Borja, A., Bouchez, A., Cordier, T., Domaizon, I., Feio, M. J., Filipe, A. F., Fornaroli, R., Graf, W., Herder, J., van der Hoorn, B., Iwan Jones, J., Sagova-Mareckova, M., Moritz, C., … Kahlert, M. (2018). The future of biotic indices in the ecogenomic era: Integrating (e)DNA metabarcoding in biological assessment of aquatic ecosystems. Sci Total Environ, 637–638, 1295–1310. https://doi.org/10.1016/j.scitotenv.2018.05.002

Sagova-Mareckova, M., Bouchez, A., Boenigk, J., Cermakova, K., Chonova, T., Cordier, T., Eisendle, U., Elersek, T., Fazi, S., Fleituch, T., Frühe, L., Gajdosova, M., Graupner, N., Haegerbaeumer, A., Kelly, A.-M., Kopecky, J., Leese, F., Nõges, P., Orlic, S., … Stoeck, T. (2021). Expanding ecological assessment by integrating microorganisms into routine freshwater biomonitoring. Water Research, 191, 116767. https://doi.org/10.1016/j.watres.2020.116767

Experimental design

Clear questions, very timely topic. Major strength is the use of eDNA samples across a whole country using a tree-of-life eDNA metabarcoding analysis (8 assays). But I see 4 aspects where additional information / analyses are urgently required to better follow the conclusions drawn and to highlight also limitations of the study better. Points 1-2 are points that need to be raised better in the discussion. How can samples size / heterogenous timing influence the results? Point 3 = better documentation / FAIR principles. Point 4 is an analysis I really would like to see given the broad geographic scale / population structure you probably witness in the data set. Here the 4 points:

1) Sample size
Key limitation, and this should be put out a little more is the number of stream sites for an index development of calibration study. This is especially true that there is much more than hard vs softbottom streams. While I’m no major expert with respect to NZ stream, in central Europe we define references and ecological quality state boundaries for over 30 different types (small, medium, large, sandy, silica-rich hard bottom, carbonate-rich hard bottom etc. pp.). This aspect needs to be discussed more carefully. There are only 53 stream sites (actually only 40 for direct calibration with historical data).

2) Sampling time
Sampling covered a very broad time frame and it is known that the detection of taxa is extremely depending on season. This is a reason why sampling for ecological quality status is restricted to a very narrow time period per year. Please elaborate how this could have impacted your results, Several eDNA from water studies address this seasonality issue. We have recently looked at the evidence ourselves. You may find the results of this preprint and the literature cited relevant. https://www.authorea.com/users/602937/articles/633564-environmental-dna-time-series-analysis-of-a-temperate-stream-reveals-distinct-seasonal-community-and-functional-shifts-but-no-influence-of-within-stream-sampling-position

3) FAIR data
A concern I have is also with respect to used reference data. Please provide all data, also provide much more information on the negative controls - how clean were they? How were samples processed? In particular how did you deal with reads in negative controls? How many reads per sample, even / uneven sequencing depths? Provide evidence (I cannot see the Smith et al. in prep. evidence). Please also provide reference data used for taxonomic assignment according to FAIR principles.

4) OTU vs. ASV
An aspect that I needs to be better addressed is: Why did you not go for OTUs (L301ff)? As for such a broad geographic gradient there can be strong phylogeographic structure within species I assume that there can be (naturally) different ASVs of the same (sensitive) species in different regions. Crustaceans that are sensitive to pollution might not be able to move between North and South of NZ and probably even at very small scales. By assigning indicator values to ASVs you might lose the ability to utilise these information. Could you at least test the TICI approach for both (ASV and OTU) approaches?
Insights on super strong population structure within species from Central Europe e.g.:
https://bmcecolevol.biomedcentral.com/articles/10.1186/s12862-016-0723-z
https://www.journals.uchicago.edu/doi/full/10.1086/674536

Validity of the findings

The study addresses a very timely topic: Can we use eDNA-based data to assess ecosystem health without relying specifically on traditional indicator taxa assignments?The chosen tree-of-life approach is one of the best and most holistic approaches I found in the current metabarcoding literature! I find the country-wide sampling amazing. I’m convinced by the consistency of the replicates analysed as well as the correlation between the historic 5-year median and the DNA data and the eDNA taxonomic and the eDNA taxonomy free MCI data (but see comments on FAIR data above).
I applaud on the careful extension of the approach you make in L527ff. It is important to consider this important general issue in biomonitoring. So in general: Congrats on the efforts, such studies are urgently needed. Conclusions seem valid, yet in view of sampling size and several other aspects raised above (2) I would advise to provide additional information, discuss aspects more carefully, especially given the low n for a whole country with a huge diversity of stream types and stressor types.

Additional comments

Minor points:
Introduce the reason and context of the Chessman interative process in the introduction.

Terminology: Sequence vs. ASV. Often in the figures (e.g. Fig. 4 and 5, S2.1) it states “sequences” - and this term is ambiguous. Do you mean reads or ASVs (I think the latter). Clarify.

I found it interesting you redesigned the Vamos et al. reverse primer we developed. Probably very good reasons for doing so - but please elaborate on the reason and how you evaluated that it was successful.

L477: Cost estimate is interesting and a nice add-on. I seems realistic too me. Please provide more information how you come to the numbers (cost break-down).

Last - many aspects you discuss are part of publications of a European project, where the introduction of DNA and eDNA-based methods atm also takes place. You might link to the discussions from these 2018ff papers, as many of the aspects you address are already found there in detail. Linking efforts across the globe makes a lot of sense.

---

## Round 0.2 · Minor Revisions

We are almost there. Florian as reviewer 2 identified a single content-related point to be considered in the discussion and some editorial things. I look forward to the last revision and the final acceptance of your paper.

·

Basic reporting

Thanks for a very good revision. I also saw and checked raw data. Looks all very professional.
Apologies that I will provide minor comments only in section 4.

Experimental design

see below

Validity of the findings

see below

Additional comments

L49 - "taxon-free" should probably mean “taxon-independent”

Long but imho very helpful and appropriate introduction

Materials and methos: Write a bit more in detail where and how you followed specific eDNA-lab procedures to avoid contamination. I agree that field blank are not a requirement if encapsulated filters are used (suggestion in CEN eDNA standard as well). However, how did you deal with reads in negative controls? Please elaborate? I suggest to substract the number of reads in negative controls from all other samples. I found that there are e.g. galaxiid sequences in the negative controls.

L219 and below: 60 cc = 60 mL? I would prefer writing the mL unit
L251 space after number
L253 delete “.” after mg
L325f: What is a “I initial site conditioning ranking”? (also L350)
L434: couldn’t -> could not

Discussion: Please rewrite the first sentence. It is too repetitive. And I strongly suggest also to remove statements like “minimal expertise”. What we observe more and more often in ringtest or also such comparisons (sorry article in German, but you’ll understand the results without that: https://www.researchgate.net/publication/371417500_Metabarcoding_versus_morphologische_Identifizierung_der_Herausforderung_gewachsen_Entomologische_Zeitschrift_Schwanfeld_133_2_2023) is that there is a tremendous variation among labs. Many struggle producing good data. It is thus not true that only minimal expertise is needed. I would suggest that you emphasize the need for clear QA/QC schemes in order to make the method living up to the expectations!
L461: Check species name (12eminate…)
L480: delete parenthesis (1st)

I did not read through all sections of the discussion, which I found good already in the first version.

---

## Round 0.3 · accepted · Accept

Thank you for the revision of the manuscript. I hereby certify that you have adequately taken into account all of the reviewers' comments, as I have checked by my own assessment of your revised manuscript. Based on my assessment as an Academic Editor, your manuscript is now ready for publication.